# Hormonal Crosstalk and Root Suberization for Drought Stress Tolerance in Plants

**DOI:** 10.3390/biom12060811

**Published:** 2022-06-09

**Authors:** Gaeun Kim, Hojin Ryu, Jwakyung Sung

**Affiliations:** 1Department of Crop Science, Chungbuk National University, Cheong-ju 28644, Korea; lkn745@chungbuk.ac.kr; 2Department of Biology, Chungbuk National University, Cheong-ju 28644, Korea; 3Department of Biological Sciences and Biotechnology, Chungbuk National University, Cheong-ju 28644, Korea

**Keywords:** root, suberization, hormones, sugar, drought

## Abstract

Higher plants in terrestrial environments face to numerous unpredictable environmental challenges, which lead to a significant impact on plant growth and development. In particular, the climate change caused by global warming is causing drought stress and rapid desertification in agricultural fields. Many scientific advances have been achieved to solve these problems for agricultural and plant ecosystems. In this review, we handled recent advances in our understanding of the physiological changes and strategies for plants undergoing drought stress. The activation of ABA synthesis and signaling pathways by drought stress regulates root development via the formation of complicated signaling networks with auxin, cytokinin, and ethylene signaling. An abundance of intrinsic soluble sugar, especially trehalose-6-phosphate, promotes the SnRK-mediated stress-resistance mechanism. Suberin deposition in the root endodermis is a physical barrier that regulates the influx/efflux of water and nutrients through complex hormonal and metabolic networks, and suberization is essential for drought-stressed plants to survive. It is highly anticipated that this work will contribute to the reproduction and productivity improvements of drought-resistant crops in the future.

## 1. Introduction

Plants with a sessile life cycle are frequently exposed to vulnerable environmental change stresses such as drought, high temperature, and light stress. Among them, drought is one of the primary environmental factors that decreases crop productivity and is the most important factor by which plants adapt to the terrestrial environment [1]. Drought is also known as one of the adverse environmental conditions that affect crop productivity [2]. Drought stress reduces soil moisture, which decreases the amount of available water for favorable plant growth, thereby hindering their growth and survival. Plants respond to drought stress by inducing molecular, physiological, and biological responses that can rapidly induce morphological and genetic changes in plants and thereby enhance a developmental plasticity [3,4].

The mechanisms of drought tolerance in plants can be divided into two main categories. First, plants seek available water by taking root deep in the soil to compensate for the lack of moisture. Under this mechanism, drought stress conditions normally enhance plant root growth and development [5]. In general, it was also reported that drought-sensitive plants tend to increase in root length and surface area to drought stress [6]. Second, plants use strategies to prevent the release of absorbed moisture from the soil. Drought stress prevents transpiration in leaves by rapidly synthesizing abscisic acid (ABA) in roots and closing the stomata through downstream signaling pathways [7]. Stomata closure via drought stress causes a decrease in CO_2_ uptake, leading to the reduction in photosynthetic efficiency [8]. It was also reported that the development of the suberin lamellar layer formed through the accumulation of suberin during drought stress increased in the root endodermal cell wall of rice [9]. In wheat (*Triticum aestivum* L.), it was confirmed that the expression of transcription factors related to root-cell-wall biosynthesis is upregulated under drought stress [10]. To date, many studies have been conducted to establish signal transduction systems for mechanisms of plant stress tolerance, such as the breeding of drought-resistant crops and increased water-use efficiency of crops. These studies may contribute to future agricultural productivity improvements [11].

In this review, we focused on recent research progress related to the ecological, molecular, and genetic approaches involved in root growth and development among various mechanisms for enhancing plant resistance to drought stress. We focus on recent advances in our understanding of the process of hormonal regulation, the relevant interactions, and types of soluble sugar metabolism in roots under drought stress. Finally, we discuss how signaling integration for root suberization can enhance drought stress tolerance through lateral root development.

## 2. Hormonal Regulation in Drought Stress

Drought stress triggered phytohormones-mediated signaling responses. Here, we evaluate the genes in relation to drought and root development in the four representative hormones (abscisic acid, cytokinin, ethylene, auxin) and highlight the interesting discovery of genes that modulate tolerance responses to drought stress (summarized in Table 1 and Figure 1).

### 2.1. ABA-Biosynthesis and Signaling 

Under drought stress conditions, abscisic acid (ABA) is rapidly synthesized in roots through the induction of ABA biosynthesis-related genes [30,31]. ABA synthesis is carried out by several enzymes such as zeaxanthin epoxidase (ZEP), ABA DEFICIENT 4 (ABA4), and 9-cis-epoxycarotenoid dioxygenase (NCED) that convert zeaxanthin to xanthoxin through oxidation in the plastid [35]. In turn, the xanthoxin present in the cytoplasm is synthesized into ABA by three enzymes; ABA DEFICIENT 2 (ABA2), ABA DEFICIENT 3 (ABA3), and ABA-aldehyde oxidase 3 (AAO3) [36]. NCED is an essential enzyme for ABA biosynthesis [30], and it was reported that *NCED3* plays an important role in ABA biosynthesis in Arabidopsis [12]. 

Drought stress signaling is also regulated by ABA-dependent and ABA-independent pathways. Both pathways are activated in drought stress and regulate the expression of various drought-inducing genes [37,38]. Recent studies have shown the complicated and close interplay between ABA-dependent and ABA-independent pathways [38,39]. ABA is recognized by PYRABACTIN RESISTANCE1 (PYR1)/PYR1-LIKE (PYL)/REGULATORY COMPONENTS OF ABA RECEPTORS (RCAR) (ABA receptors). The activated ABA receptors directly inhibit protein phosphatase Protein phosphatase 2C (PP2C) activity. As a result, subclass III SNF1-RELATED KINASE 2s (SnRK2s), which are negatively regulated by PP2C, are activated and gene-expression regulation of ABA signaling related pathways [40]. ABA-modulating gene expression is regulated by the ABA-responsive element-binding proteins (AREB)/ABA binding factor (ABFs) and basic leucine zipper (bZIP) transcription factors, which are phosphorylated by SnRK2s. Consistently, overexpressing plant of the *TaSnRK2.4* of wheat (*Triticum aestivum* L.) was reported to have longer basal roots under normal conditions and to enhance drought tolerance in drought stress [13]. The *areb1 areb2 abf3* triple knockout mutant increased sensitivity to drought stress and was insensitive to primary root growth inhibition via ABA [14]. In addition, the ABF/AREB transcription factor plays an important role in regulating the drought response by directly interacting with the dehydration-responsive element-binding protein (DREB) 2A, DREB1A, and DREB2C, which are involved in ABA-independent pathways [15]. In addition to ABRE-mediated gene regulation, myeloblastosis (MYB) and MYC Proto-Oncogene (MYC) transcription factors are also involved in ABA-mediated gene regulations [41]. *AtMYB96*, an up-regulator of ARFs, is induced by ABA signaling under drought stress and acts as a negative regulator of lateral root development by increasing the expression of *ABA-INSENSITIVE 5* (*ABI5*) [42]. Among the transcription factors activated by SnRK2 in rice, *OsNAC10* is mainly expressed in roots and induced by drought and ABA [16]. Consistently, the overexpression of *OsNAC10* in roots was found to increase root development and improve the drought tolerance of plants. The ABA-independent pathway regulates the expression of ethylene response factor (ERF) genes in response to drought stress in rice and wheat. This pathway is known to enhance the drought tolerance of crops by promoting root development [33].

### 2.2. Cytokinin (CK) Signaling 

Cytokinin (CK) is mainly biosynthesized in roots and known to be involved in cell division, meristem identification, the inhibition of senescence, and enhancing sink activity. Recent studies have shown that CK synthesis and signaling pathways respond immediately to drought stress [43]. CK plays critical roles in root formation and morphogenesis through antagonistic interactions with Auxin [44,45]. CK is directly recognized by the CK receptor Arabidopsis histidine kinases (AHKs) at the plasma membrane. Upon the binding of CK to AHK proteins, the receptors activate their downstream signaling cues via a two-component system phosphor relay cascade with histidine-containing phosphor-transfer proteins (AHPs). The phosphate group transmitted from the AHKs is transferred to the receiver domain of A or B-type response regulators (ARRs) through AHPs. Phosphorylated B-type ARRs, in turn, regulate the transcription of numerous genes, including A-type ARRs. The content of CK is generally reduced in drought stress [46,47]. Cytokinin oxidase (CKX) plays a critical role in regulating the homeostasis of CK in plants. Consistently, overexpressing *CKX* genes in plants promoted primary root length and lateral root formation under drought stress [48]. Furthermore, *ahk2 ahk3* double knock-out mutants were more resistant to dehydration than wild-type plants, indicating that cytokinin is a negative regulator in drought stress [17]. AHP2, AHP3, and AHP5 act as negative regulators of drought stress [18], and the expression of CK-responsive regulators ARR1, ARR10, and ARR12 (type-B RR) is suppressed in drought stress in plants [19]. These results indicate that CK is a negative regulator of drought stress [49].

### 2.3. Auxin Signaling 

Auxin is an essential hormone for regulating organogenesis, cell division and elongation during root differentiation [50]. Auxin is also involved in the overall process of plant growth inhibition under drought stress [51]. The auxin response occurs when gene expression is regulated by the auxin-reactive element (ARE) of DNA. Transcription of these auxin-reactive genes is initiated via the auxin response factor (ARF), which regulates the ARE [52]. The Aux/IAA protein is a regulator that inhibits the function of the ARF transcription factors [53]. The auxin-binding TIR1 (transport inhibitor response 1) receptor is an E3 ubiquitin ligase that promotes degradation of Aux/IAA proteins via the 26s proteasome pathway [54]. In rice (*Oryza sativa* L.), it was confirmed that the auxin transporters PIN-FORMED (PIN) proteins OsPIN5b and OsPIN2 are upregulated by drought stress [20]. OsPIN3t, which is directly induced in response to auxin in rice, plays a key role in the root growth and development of rice and was confirmed to be involved in drought stress tolerance mechanisms [21]. In addition, IAR3 hydrolase, which hydrolyzes auxin to generate free auxin in Arabidopsis, was found to enhance the development of lateral roots by generating active auxin under drought stress [22]. Auxin is also known to regulate lateral root development by upregulating *MYB96* expression in roots [42]. Recently, it was confirmed that the *DEEPER ROOTING 1* (*DRO1*) gene, which determines the QTL involved in the root growth angle, is negatively regulated by auxin, and that *DRO1*-overexpressed rice presented higher drought tolerance [23]. Plant microRNA miR393 was confirmed to be involved in the adaptation of roots to drought conditions through the attenuation of auxin signaling outputs [24]. However, more systematic studies are still needed to determine the genetic mechanism directly regulated by miR393.

### 2.4. Ethylene Signaling 

Ethylene is also one of the critical plant hormones involved in lateral root development and root growth [55]. Ethylene acts synergistically with auxin in the regulation of primary root elongation [55,56]. It was confirmed that the biosynthesis of ethylene is increased under abiotic stresses such as drought [57]. Ethylene is recognized in the ER membrane by five receptors: ETHYLENE RECEPTOR1 (ETR1); ETHYLENE RESPONSE SENSOR1 (ERS1); and the subfamily II proteins ETR2, ERS2, and ETYLENE INSENSITIVE4 (EIN4) [58]. The downstream signaling responses are proceeded through the CONSTITUTIVE TRIPLE RESPONSE (CTR1), EIN2, EIN3/ETHYLENE-INSENSITIVE3-LIKE 1 (EIL1), and ETHYLENE RESPONSE FACTORs (ERFs) [59]. Under drought stress conditions, the expression of *ERFs* is induced. Then, the ERFs bind to a dehydration-responsive element (DRE) to activate specific stress-responsive genes [60]. In Arabidopsis, *AtERF1*, *AtERF5* and AtERF6 genes were highly induced by drought condition [25,26]. *OsERF48-* and *OsERF71-*overexpressing rice plants showed improved root growth and drought tolerance [27,28]. In addition, drought stress resistance was reported to correspond with an increase in root length and biomass in plants overexpressing *TSRF1* [29].

## 3. Soluble Sugars in Drought and Sugar-Responsive Metabolism

Soluble sugars in plants, such as sucrose, glucose, and fructose, are broadly used as metabolic energy sources, cellular structural components, and regulators of plant growth and development processes [61,62,63]. These sugars also serve as negative regulators of plant responses to biotic/abiotic stresses [64]. Drought tends to accumulate the level of soluble sugars in the roots (Figure 2) of Arabidopsis [65,66], maize [67], and rice [68]. In particular, monosaccharides including glucose, fructose, and trehalose are abundant in rice [69], Medicago [47], and Prunus [70] as a consequence of drought.

Perturbations in the cellular levels of soluble sugars under abiotic stresses, including drought, induce signal transduction closely associated with endogenous hormones and thus broadly regulate biochemical and molecular metabolism from the cellular to plant level (Figure 2) [75,76]. Trehalose, reported as a tolerant factor against abiotic stresses, is biosynthesized from trehalose-6-phosphate (T6P) via biochemical reactions through trehalose-6-phosphate synthase (TPS) and trehalose phosphate phosphatase (TPP) [77]. T6P, an essential moderator for sugar metabolism in plants [71], is an indicator of ABA signaling-responsive snf1-related kinase (SnRK1) activity and play a key role in modulating cellular sucrose homeostasis via negative feedback to sucrose [72]. Indeed, the level of trehalose was preferentially enhanced by drought stress in several crop plants including rice and soybean, and these responses were commonly observed in wild type and mutant (drought-resistant) lines [69,70,73,78,79,80,81,82]. Under stress conditions, SnRK1, a sensory protein that regulates gene expression in relation to cellular energy levels, not only regulates plant development and signal perception, down-stream reactions, and cellular homeostasis [83] but also acts in an opposing way against target of rapamycin (TOR), a crucial protein in plant growth and development [84]. The TORC1 (RAPTOR-LST8-TOR complex), which is activated due to inactivation of the PP2C-mediated SnRK1/2 complex, induces inactivation of PYRs, an ABA receptor, via phosphorylation and leads to root growth under favorable plant growth conditions [85]. In contrast, drought stress enhances the cellular levels of sugars such as sucrose, T6P, and trehalose. Notably, T6P restricts the activation of SnRK1 via phosphorylation [86]. Phosphorylated SnRK1/2 promotes the breakdown of the RAPTOR-TOR-LST8 complex via the phosphorylation of RAPTOR, and greater ABA-mediated activation of the PYR–PP2C complex promotes the activation of SnRK2, which triggers a stress response [87,88]. In a previous study, greater accumulation of trehalose led to the simultaneous regulation of root growth and stomata closure through ABA-response factor 2 (ARF2) in Arabidopsis [89] and strengthened drought tolerance in OsTPP3-overexpression rice [73]. In terms of the regulatory role of OsTPS8 in rice plants, the knock-out of OsTPS3 was found to be insensitive to ABA with a relatively lower level of soluble sugars. In contrast, over-expression accelerated the deposition of suberin in rice roots [74]. These observations suggest that root growth under drought is strongly regulated by ABA-responsive trehalose-triggered metabolism. Thus, T6P, TPP, and trehalose are indispensable molecules that enhance tolerance and/or resistance to drought. Figure 2 summarizes the metabolic regulation by trehalose and ABA against drought condition. 

## 4. Suberin Biosynthesis in Plant Roots and Drought-Derived Modification

Suberin is defined as a glycerol-based aliphatic polyester complex connected with cross-linked polyaromatics and waxes [90]. The suberin precursor is synthesized via enzymatic reactions of the CYP86 subfamily of P450 monooxygenase, acyl-CoA synthetases of the LACS family, and acyltransferases of the FPAT family [91] and transferred toward the cell wall by the ATB-binding cassette (ABC) transporter. Among genes encoding suberin-biosynthetic enzymes, fatty acyl-CoA reductase (FAR) 1, 4, and 5 were found in Arabidopsis roots [92], and ABC transporters are directly involved in the movement of suberin polyesters in triple mutant (*abcg2*, *abcg6*, and *abcg20*) in Arabidopsis roots where structural and compositional modifications are observed [93] (Figure 3A). 

Suberization is a physical process of suberin deposition, and the suberin lamellae are a suberin-deposited layer formed between the cell wall and plasma membrane after development of the Casparian strip (CS) [94]. The suberin lamellae are considered to exert partial restriction of the endodermis [95] but function as a physical barrier restricting the transport of water and nutrients [96]; thus, suberization is not an essential process during root development [94] (Figure 3B).

Li et al. [97] reported that root suberization is closely associated with CS-synthetic gene groups, including SHORT-ROOT (SHR), SCARECROW (SCR), MYB36, SCHENGEN (SGN) 3, SGN4, ENHANCED SUBERIN 1 (ESB1), Peroxidases (PODs), Casparian strip membrane domain proteins (CASPs), and Casparian strip integrity factor (CIF)1/2 (Figure 3B and Figure 4F). Enhanced endodermal suberin in *esb1-1* is driven by the malfunction of ESB1 in Arabidopsis roots, and compositional modification between CS and suberin leads to the regulatory function of suberin in water and solute transport [98]. 

The formation of the suberin lamella is mediated by ABA-signaling transduction. The SHR and MYB36 [97,99], which promote endodermal differentiation in roots, activate enhanced biosynthesis of the suberin lamella and interact with the ABA-responsive pathway [100]. These genes are directly regulated and/or mediated by *SCR* [101]. The ABA-dependent activation of *SHR* and *SCR* under drought conditions promotes a transcription factor, miR165/166, in the root endodermis and decreases the level of polyhydroxyvalerate (PHB) [102]. The MYB39 involved closely in Casparian strip formation is linked to a promoter of FAR5, which is associated with suberin biosynthesis and deposits suberin in the root endodermis [100,103]. Moreover, MYB41, MYB53, MYB92, and MYB93 are essential regulation factors during root suberization [104] (Figure 3B).

Recently, it has been reported that ABA as well as other phytohormones are involved in suberin biosynthesis and suberin lamellae formation. The synthesis and degradation of suberin is greatly dependent upon GDSL-type esterase/lipase (GELP) mediated by auxin (Figure 3B,C), and endodermal suberization is closely coupled with the differentiation and growth of lateral roots [105] (Figure 3C). Sam et al. reported that suberization was induced in the endodermis of auxin-treated roots [106]. This is closely related to the aforementioned involvement of ABA in suberization. In addition, it has been reported that gibberellin and ABA accumulate in the root endodermis via NPF2.14 acting as a subcellular transporter, and the accumulated hormones regulate suberization [107]. However, the regulatory network through in which auxin and GA control suberization remains largely unknown. Additionally, it is not clear whether other phytohormones are related to suberization.

The relationship between suberization and drought or osmotic stress has been elucidated. Drought not only triggers ABA-signal transduction but also promotes trehalose biosynthetic genes and suberin deposition [109]. The formation of the suberin lamellae via ABA-signaling metabolism was observed in drought-stressed rice [10] and barley plants [108]. In addition, the ABA-treated wheat observed an improvement of suberin lamellae in root. Arabidopsis accelerated suberin deposition in drought, indicating that drought induces suberin biosynthesis [110]. Additionally, the suberin limit uncontrolled Na uptake in high-salinity stress in arabidopsis [111]. These results suggest that suberin deposition is one of the adaptation strategies of plants that controls the cellular level of water and nutrients under drought stress. Recently, in rice, profiling of genes involved in the regulation of root development under various moisture conditions including drought was performed [112]. As with previous findings, the water-deficit leads to some key physiological responses in suberin synthesis and the same trend can be confirmed in suberin staining result. However, it has not been revealed whether phytohormones other than ABA are involved in suberization in drought stress. Based on the previous involvement of ethylene and GA in suberization, it is necessary to examine whether the corresponding hormones are affected by suberization and development of entourage in drought stress. In addition, studies on the relationship between phytohormones (CK, ethylene, etc.) and suberization, which have not yet been elucidated, are needed.

## 5. Conclusions and Future Perspectives

The increasing global population requires that agricultural production should be raised by at least 50% compared to current statistics. Nevertheless, diverse environmental stresses due to climate change are threatening sustainable food production systems. Therefore, it is imperative to explore the complex network between sugars and hormones in physiological responses, and to provide clues to enhance plant adaptation against accelerating global drought events. The plant hormone ABA plays an important role in regulating the expression of numerous genes in plants to respond to drought and increase tolerance for survival. Under drought stress, the ABA hormone is confirmed to be involved in root development through interactions with various hormones and signaling [113,114] (Figure 4). The antagonistic interplay between CK and ABA in response to drought stress is caused by indirect or direct interactions between signaling genes [32]. Drought stress directly phosphorylates ARR5 (type-A RR5), a negative regulator of CK signaling, through the activation of SnRK2s, a component of ABA signaling in plants, thereby inhibiting CK activity [32] (Figure 4C). In addition, type-B RRs including *ARR1*, *ARR10*, and *ARR12* genes, which are known to be suppressed under drought, are all suppressed by ABA [19]. Type-B RR inhibits lateral root development [115], and ARR1 and ARR12 inhibit *PIN* expression by activating the IAA3/SHORT HYPOCOTYL 2 (SHY2) involved in auxin signaling [116]. The auxin signal required for lateral root development is formed in the root endodermis via IAA3/SHY2 [117], and ARF is expressed and plays a role in lateral root development, initiation, and inhibition [118]. It is known that the activity of ARF is affected by ABA under drought stress. In rice, miR167, which targets ARFs, is suppressed by ABA [119]. In drought stress, miR167 promotes lateral root formation by regulating ARF6 and ARF8, and miR160 regulates the expression of ARF10, ARF16, and ARF17 to affect lateral root formation and root length [120]. MYB96, an upstream regulator of ARFs, is affected by ABA and auxin. MYB96’s expression is induced in drought stress and involved in lateral root formation [42,121] (Figure 4B,D). In addition, during the process of activating the lateral root meristem, the expression of the ABA signaling transcription factor ABI5 is regulated by ARFs activated by MYB96 [122,123] (Figure 4D). *ABI5* expression is mainly observed at the root tip, and the length growth of the lateral root is inhibited as the expression is activated by the MYB96-ARF module [42]. It is well described that soluble sugars including sucrose and trehalose play essential roles as a signal molecule with an orchestration of endogenous hormone-mediated responses against drought stress [66,71]. In particular, trehalose-6-phosphate plays an elicitor to manipulate drought-mediated metabolic cascades through SnRK protein activation [71]. The suberization, a physical barrier located in endodermis, is deposited through fatty acid synthetic-downstream reactions promoted by an ABA-responsive transcription factor, MYB39 [100,103].

In this review, we discussed how hormonal signaling pathways and soluble sugar metabolism are linked with increased drought stress resistance in plants. We also explored how they play a critical role in the root endodermis suberization. However, research into how this complex signal transduction network aided natural evolution or domesticated through artificial selection by adjusting to environmental changes is still unclear. Climate change, exacerbated by global warming, poses numerous difficulties to humanity’s ability to sustain food security. It is expected that accurate understanding of how the different genetic processes involved in plant adaptation to the ever-changing environment interact will be attainable, and that responses will be achievable using the sophisticated and advancing genetic engineering techniques. 

## Figures and Tables

**Figure 1 biomolecules-12-00811-f001:**
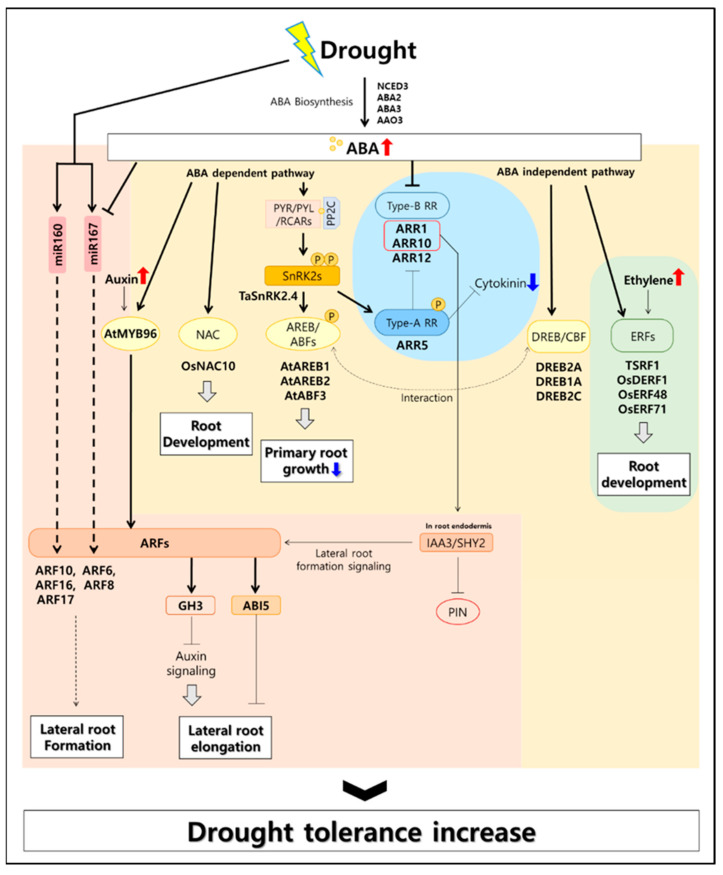
Root-associated hormonal signaling model for drought stress tolerance. In drought, plants induce abscisic acid (ABA) biosynthesis genes to produce large amounts of ABA in plant roots [30,31]. ABA synthesized in large quantities is directly involved in the drought-tolerant metabolism of plants through the process of ABA signaling. ABA activates the ARF transcription factor related to auxin in response to increased drought tolerance in roots, and cytokinin antagonizes ABA [32]. In addition, the ABA-independent pathway activates the ethylene signal transduction transcription factor to help the root development [33,34].

**Figure 2 biomolecules-12-00811-f002:**
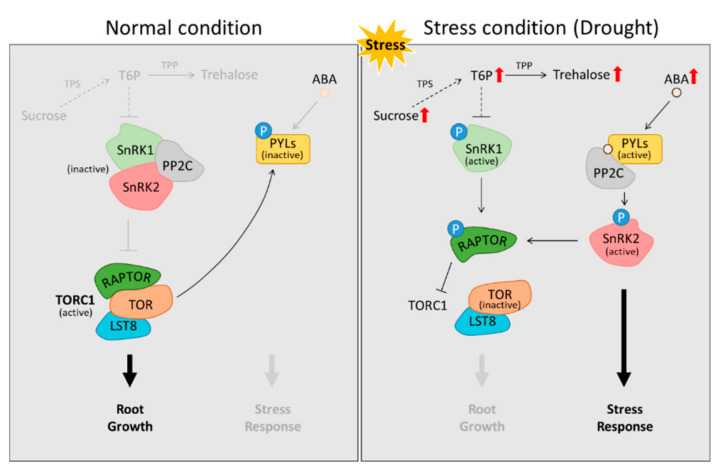
Metabolic regulation by trehalose and ABA against drought stress. The activation of SnRK1/2 by trehalose-6-phosphate and ABA breaks down the RAPTOR-TOR-LST8 complex and triggers a stress response. Drought tends to accumulate the level of soluble sugars in the roots (Red arrow) [65,66,71,72,73,74]. The LATERAL ORGAN BOUNDARIES DOMAIN transcription factors, LST8; PP2C, Protein phosphatase 2C; PYL, pyrabactin resistance-like; RAPTOR, REGULATORY-ASSOCIATED PROTEIN OF TOR; SnRKs, SNF1-RELATED KINASE; T6P, Trehalose-6-phosphate; TOR, TARGET OF RAPAMYCIN; TPP, T6P-phosphatase; TPPE, Probable trehalose-phosphate phosphatase E; TPS, trehalose-6-phosphate synthase.

**Figure 3 biomolecules-12-00811-f003:**
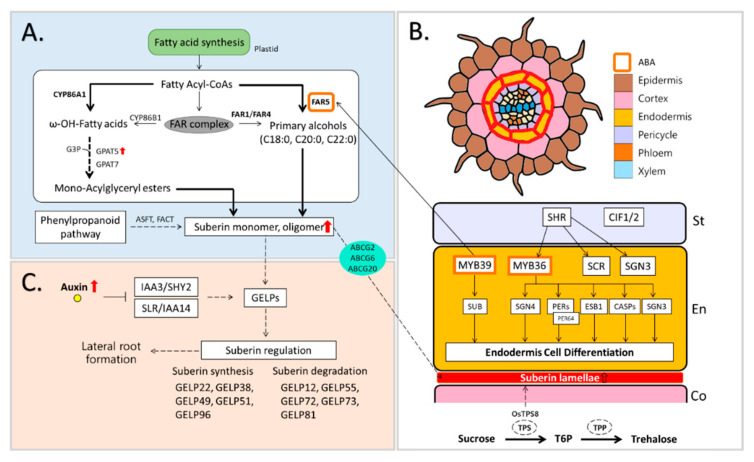
Suberin-related metabolism in drought-associated roots. (**A**) Suberin biosynthesis [95] (**B**) Interaction between Casparian strip-related genes for the formation of suberin lamellae [9,97,108]; (**C**) synthesis and degradation of suberin by auxin-mediated GELPs [105]. ABCG, ATP-binding cassette (ABC) transporters; ASFT, Aliphatic suberin feruloyl transferase; CASPs, Casparian strip membrane domain proteins; CIF, Casparian strip integrity factor; ESB1, enhanced suberin 1; FACT, Aliphatic suberin feruloyl transferase; FARs, fatty acyl reductases; GELPs, GDSL-type esterase/lipase; GPATs, Glycerol-3-phosphate acyltransferase; IAAs, Indole-3-acetic acid; SCR, SCARECROW; SGN, SCHENGEN; SHR, SHORT-ROOT; SHY2, IAA3; SLR, IAA14; SnRKs, SNF1-RELATED KINASE; T6P, Trehalose-6-phosphate; TPP, T6P-phosphatase; TPS, trehalose-6-phosphate synthase.

**Figure 4 biomolecules-12-00811-f004:**
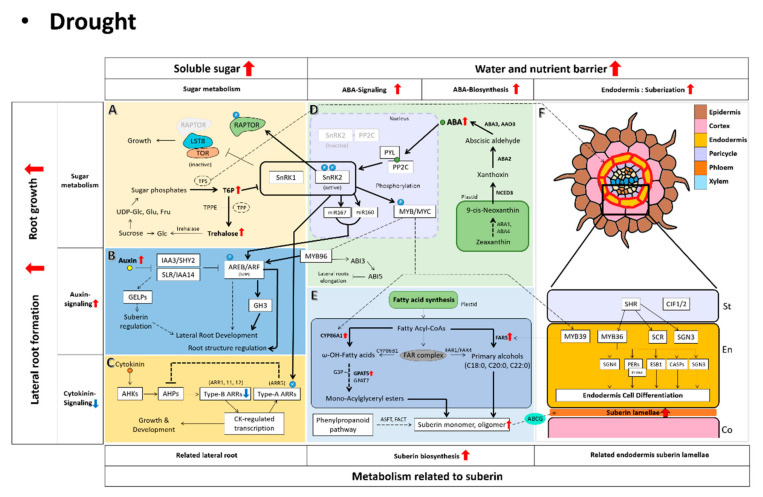
Root signaling model in drought stress. In drought, root-resistant signaling is carried out based on ABA signaling, and trehalose and sucrose sugars are also seen as major indicators of drought stress. The phenomenon of drought tolerance directly in the roots is seen as the formation of suberization and development of the roots. The figure shows the relationship between the drought-tolerant signal and the resulting phenotype. The red arrows indicate an increase in drought stress. The bold arrows indicate increased signaling in drought stress. (**A**) Plants increase sugar levels under stress conditions such as drought [65,66]. Among them, trehalose-6-phosphate(T6P) plays a key role in regulating sugar metabolism in plants [71]. SnRK1 related to ABA signaling uses T6P as a measure of cellular energy status, and T6P contributes to intracellular sucrose homeostasis by functioning as a negative regulator of sucrose [72]. By regulating TPS, the sensitivity of ABA and suberin are also regulated, thereby regulating the suberization of roots and providing stress tolerance [73,74]. Based on this mechanism, the metabolism of sugars related to trehalose and sucrose can be an indicator of stress tolerance metabolism in plants. (**B**) Auxin regulates lateral root development by upregulating the expression of the ABA signaling factor MYB96 [42]. The appearance of lateral roots affects the synthesis and degradation of suberins depending on the type of GELP lipase affected by auxin 110 [105]. Based on these identified metabolic pathways, it is necessary to examine the process of auxin signaling in relation to plant-entourage development in drought. (**C**) In drought, cytokinin content generally decreases [46,47]. Genetic studies of Arabidopsis thaliana support that cytokinin is a negative regulator in drought stress [49]. (**D**) In drought, plants induce abscisic acid (ABA) biosynthesis genes to produce large amounts of ABA in plant roots [30,31]. ABA synthesized in large quantities is directly involved in the drought-tolerant metabolism of plants through the process of ABA signaling. (**E**) This figure presents the biosynthetic metabolism of suberin. In drought, the metabolism of suberin biosynthesis increases overall. It was confirmed that the suberin synthesis process was influenced by ABA, a hormone that occurs frequently in drought [95]. (**F**) The formation of suberin lamellae is caused by the interaction of Casparian strip-related genes, ABA hormones, and suberin-related genes, which are the previous developmental stages [97]. A suberin lamellae layer was formed at the roots to prevent the release of nutrients in drought [9], which was also observed in the osmotic [108] and salt stress [95] related to moisture.

**Table 1 biomolecules-12-00811-t001:** Contribution of phytohormones-responsive genes to drought tolerance in plants.

Related	Genes	Species	Genetic Manipulation	Effect on Tolerance	Ref.
ABA	*AtNCED3*	Arabidopsis	Over-expression	Transgenic plants were more resistant to drought stress than WT.	[12]
ABA	*TaSnRK2.4*	Arabidopsis	Over-expression	Under normal conditions, the primary root lengthens.	[13]
ABA	*AtAREB1*, *AtAREB2*, *AtABF3*	Arabidopsis	Knock-out	Transgenic plants were more resistant to ABA compared to primary root growth and displayed reduced drought tolerance.	[14]
ABA	AtDREB2A, AtDREB1A, AtDREB2C	Arabidopsis	-	ABA-independent proteins (DREB2A, DREB1A, and DREB2C) interact with each other and play an important role in regulating drought response.	[15]
ABA	*OsNAC10*	O. sativa	Over-expression	Transgenic plants increase root development and improve the drought tolerance of plants.	[16]
CK	*AtAHK2* *AtAHK3*	Arabidopsis	Knock-out	Transgenic plants more resistant to dehydration than wild-type plants	[17]
CK	*AtAHP2* *AtAHP3* *AtAHP5*	Arabidopsis	Knock-out	AHP2, AHP3, and AHP5 responses to drought stress in a negative and redundant manner.	[18]
CK	*AtARR1* *AtARR10* *AtARR12*	Arabidopsis	Knock-out	Triple mutant showed a significant increase in drought tolerance versus WT.	[19]
Auxin	OsPIN2	O. sativa	-	Induced by drought.	[20]
Auxin	OsPIN5b	O. sativa	-	Induced by drought.	[20]
Auxin	OsPIN3t	O. sativa	Over-expression	Transgenic plants improved drought tolerance and led to root development.	[21]
		O. sativa	Knock-out	Transgenic plants resulted in slightly shorter adventitious roots.	[21]
Auxin	*AtIAR3*	Arabidopsis	Knock-out	Transgenic plants were significantly more sensitive to drought than WT and formed fewer lateral roots.	[22]
Auxin	*DRO1*	O. sativa	Over-expression	Transgenic plants have higher drought tolerance.	[23]
Auxin	miR393	Arabidopsis	-	miR393-mediated attenuation of auxin signaling is essential for the inhibition of lateral root growth by ABA or osmotic stress.	[24]
Ethlyene	*AtERF1*	Arabidopsis thaliana.	Over-expression	*ERF1* could enhance drought survival.	[25]
Ethlyene	*AtERF5* *AtERF6*	Arabidopsis thaliana.	Knock-out	Double mutants grow better under osmotic stress.	[26]
Ethlyene	*OsERF48*	O. sativa	Over-expression	Transgenic plants improved drought tolerance and root growth.	[27]
Ethlyene	*OsERF71*	O. sativa	Over-expression	Transgenic plant enhanced drought tolerance by enabling root morphological adaptation	[28]
Ethlyene	*TSRF1*	O. sativa	Over-expression	Transgenic plants increase root length and weight and improve drought tolerance.	[29]

## Data Availability

Not applicable.

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
