# Peer review of "Hormonal Crosstalk and Root Suberization for Drought Stress Tolerance in Plants"

_biomolecules, 2022, doi:10.3390/biom12060811_

Round 1

Reviewer 1 Report

In this review article, the authors presented some important findings on the hormonal cross-talk and root suberization in plants under drought. Drought-induced hormonal regulation in plants have been reported by many researchers and there are some review article is also available. However, integrating hormonal cross talk and suberization is somehow new and interesting.

I suggest the authors discuss more about the actual relationship between phytohormones and suberization.

Apart from the molecular functions how they affect the physiology of plant should be discussed more. For example, osmoregulation, water relations etc.

Some of the references are old. Please cite updated references.

There are plenty of formatting errors e.g. Scientific names are not Italic.

Language should be checked by English expert.

Author Response

We are sincerely grateful you for your time and consideration on our manuscript, “Hormonal crosstalk and root suberization for drought stress tolerance in plants” to Biomolecules, (biomolecules-1745249). Thus, it is with great pleasure that we resubmit our article for further consideration. We have incorporated changes that reflect that detailed suggestions you have graciously provided. We also hope that our edits and the responses we provide below satisfactorily address all the issues and concerns of editor and the reviewers have noted.

The original referee comments are provided in black color, whereas our answer are given in blue. The appropriate changes made in the revised manuscript are highlighted.

The main problem was that the recent thesis was supplemented a little more and needed explanation and English correction. To solve this problem, we supplemented the contents of the latest papers, and reviewed the English notation and proofreading.

Sincerely,

Ga-Eun Kim

Reviewer 2 Report

For sessile plants in terrestrial environments, numerous unpredictable environmental changes could have a significant effect on plant growth and development. In particular, climate change caused by global warming is increasing drought stress in agricultural fields and causing rapid desertification. Many scientific advances have been achieved to solve these problems for agricultural and plant ecosystems. In this review paper, the authors discuss recent advances in our understanding of the physiological changes and strategies for plants undergoing drought stress and also the different hormone signals involved in this process. This review paper will helpful for the reproduction and productivity improvements of drought-resistant crops in the future. In general, this view paper is well-written and the summary and discussion of the current progress with hormone cross-talk in drought stress comprehensive. 

Minor points:

1.     There is no title for the Table 1.

2.     As a review paper, the author should cite more papers published in 2022, I could only find one reference from 2022.

3.     The application value should be discuss more in the discussion part!

Author Response

(The authors gave the same response as above.)

Reviewer 3 Report

Dear authors, attached is the PDf file with the text highlighted in yellow. These are minor corrections or sentences which need your attention. Overall, this is a good paper, well-executed, but the figures need more quality (pixels).

Author Response

(The authors gave the same response as above.)
